

# Multi-angle aerosol optical depth retrieval method based on
# improved surface reflectance
Lijuan Chen[1], Ren Wang[1], Ying Fei[2], Peng Fang[2],Yong Zha[2], Haishan Chen[1*]
[1]Key Laboratory of Meteorological Disaster, Ministry of Education (KLME)/Joint International
Research Laboratory of Climate and Environment Change (ILCEC)/Collaborative Innovation
Center on Forecast and Evaluation of Meteorological Disasters (CIC-FEMD), Nanjing University
of Information Science and Technology, Nanjing 210044, China
[2]Key Laboratory of Virtual Geographic Environment of Ministry of Education, Jiangsu Center for
Collaborative Innovation in Geographical Information Resource Development and Application,
College of Geographic Science, Nanjing Normal University, Nanjing 210023, China
Correspondence: Haishan Chen (haishan@nuist.edu.cn)
**Abstract**
Retrieval of terrestrial aerosol optical depth (AOD) has been a challenge for satellite Earth
observations, mainly due to the difficulty of estimating surface reflectance caused by
land-atmosphere coupling. Current satellite AOD retrieval products have low spatial resolution
under complex surface processes. In this study, based on our previous studies of AOD retrieval,
we further improved the estimation method of surface reflectance by establishing an error
correction model and then obtained a more accurate AOD. A lookup table is constructed using the
Second Simulation of Satellite Signal in the Solar Spectrum (6S) to obtain high-precision retrieval
of AOD. The retrieval accuracy of the algorithm is verified by AERONET (Aerosol Robotic
Network) observations. The results indicate that the retrieved AOD based on the improved method

Atmospheric
Measurement
Techniques



Discussions

of this study has advantages in fewer missing AOD pixels and finer spatial resolution, as
compared to the MODIS AOD product and our previous estimation method. Among the nine
MISR angles, the optimal correlation coefficient (R) of retrieved AOD and observed AOD can
reach 0.89. Root mean square error (RMSE) and relative mean bias (RMB) can reach a minimum
values of 0.20 and 0.32, respectively. This study will help to further improve the accuracy of
retrieving multi-angle AOD at large spatial scales and long time series.
Keywords: surface reflectance; aerosol optical depth; satellite remote sensing; MISR; MODIS
**1. Introduction**

Aerosols are liquid or solid particles suspended in the atmosphere, with particle diameters

ranging from approximately 0.001 to 100 μm (Giles et al., 2019). Aerosols have a large impact on
the Earth's radiation budget balance and the uncertainties are difficult to estimate (Holben et al.,
2001; Li et al., 2020; Berhane et al., 2021; Sun et al., 2022), thus direct and indirect effects of
aerosols have received widespread attention in the study of climate change mechanisms
(Hatzianastassiou et al., 2009; Dao et al., 2014; Daniel et al., 2014; Samset et al., 2018; Li et al.,
2018; Huang et al., 2021). In addition, concentrations of aerosols may pose a serious threat to
human health (Lee et al., 2010; Dehghani et al., 2012; Mironova et al., 2015). The parameters of
the optical properties of aerosols include aerosol optical depth (AOD), scattering phase function,
single scattering albedo and absorbing optical depth, etc. As an important parameter, AOD is
defined as the integral of aerosol extinction coefficient in the vertical direction. AOD describes the
attenuation effect of aerosols on light and also reflects an important indicator of the degree of air
pollution. Over the past two decades, multi-channel spectrometers carried by multiple



geostationary and polar orbit satellites have been used for AOD retrieval. The AOD products
obtained from satellite retrieval are widely used in the study of atmospheric environment
(Kaufman et al, 1997; Xie et al., 2019; Chen et al., 2021). Although the retrieval accuracy of AOD
is constantly improving, there is still a lot of room for improvement in the retrieval results over
land.

Scholars have conducted studies using multi angle sensors. Flowerdew et al. (1996) utilized

Along Track Scanning Radiometer 2 (ATSR-2) dual angle observation data, based on the
approximate condition of minimum variation of surface reflectance with wavelength, and using
the assumption of independent invariance of ground features and Lambertian bodies, simulated
using a bidirectional reflection radiation transfer model, and proposed a dual angle algorithm
(ATSR-DV) to invert AOD over land. Kokhanovsky et al. (2009) used the ATSR-DV algorithm to
invert the AOD over Germany on October 13, 2005, and compared the retrieval results with
MEdium Resolution Imaging Spectrometer (MERIS) and MISR products, indicating that the
ATSR-2 algorithm is also suitable for Advanced Along Track Scanning Radiometer (AATSR).
Sundstrom et al. (2012) obtained an aerosol model of eastern China based on Aerosol Robotic
Network (AERONET) observation data, and used the ATSR-DV algorithm to retrieve the
proportion of AOD and coarse to fine particles from AATSR data. Abdou et al. (2005) compared
the MISR AOD and the Moderate-resolution Imaging Spectroradiometer (MODIS) AOD products
carried by Terra using data from 62 AERONET observation sites. The results showed that over
land, the MODIS AOD in the 0.470 um and 0.660 um channels was 35% and 10% higher than
MISR. In coastal and desert areas, the MODIS retrieval error was relatively large, while over the
ocean, in the 0.470 um and 0.660 um channels, the MISR was 0.1 and 0.05 higher than the



MODIS AOD value, respectively, mainly depends on the accuracy of radiometric calibration.
Martochik et al. (1997) proposed an algorithm for extracting aerosol optical parameters using
MISR multi angle observations. The results showed that in the presence of dense vegetation over
land, AOD was extracted using its low reflectivity and multi angle observations. If dense
vegetation did not exist, AOD and aerosol models were determined using the reflectance function
spectral contrast angle dependence relationship. As a new remote sensing tool, multi angle remote
sensing has the ability to provide aerosol characteristics such as optical depth, single scattering
albedo, and phase function with sufficient precision, which is more suitable for playing its unique
role in aerosol research than traditional single angle optical remote sensing (Dubovik et al., 2019).
Multi angle remote sensing retrieval of aerosol optical properties can utilize the angle information
contained in satellite signals to better separate the contributions of the surface and atmosphere,
making it suitable for some bright surfaces. This provides a new approach for AOD retrieval.

The surface reflectance measures the ability of land acquisition to absorb and reflect solar

radiation. The surface reflectance is relatively complex on land, and its contribution is received by
satellite detectors after atmospheric scattering and absorption. Satellite observations are obtained
as a coupling of the two, making it difficult to directly distinguish between surface reflectance and
atmospheric scattering. Therefore, simultaneous retrieval of atmospheric aerosols and surface
reflectance is the goal pursued by quantitative satellite remote sensing (Deuzé et al., 2001). In
optical remote sensing, the blue band has shorter wavelengths, relatively low surface reflectivity,
and there is relatively more reflection and scattering caused by the atmosphere. Therefore, AOD is
generally retrieved through the blue band. In the process of AOD retrieval, the overestimation of
the surface reflectivity will lead to an underestimation of AOD, and the underestimation of surface



reflectance will lead to an overestimation of AOD. Separating atmospherically generated
reflectance and surface reflectance from apparent reflectance is one of the difficulties of AOD
retrieval (apparent reflectance is the reflectance at the top of the atmosphere). In general, signals
such as aerosols are weaker compared to surface signals (Dong et al., 2023). Previous studies have
shown that when using satellite remote sensing to retrieve AOD, an intercept error of 0.01 in
surface reflectance can result in an retrieval error of approximately 0.1 (Zhang et al., 2021).
Therefore, accurate estimates of surface reflectance are an important basis for aerosol retrieval.

A high-precision AOD product obtained from retrieval is of great significance for monitoring

changes in atmospheric pollution and providing decision-making for pollution control. Observing
the spatial distribution of AOD is very important for daily monitoring of air pollution. In addition,
aerosol particles can affect the energy balance between the land and the atmosphere by absorbing
and scattering solar radiation, thus affecting the global climate system. To further improve the
retrieval accuracy and resolution of AOD, this study uses data from nine camera angles in the blue
band of MISR L1B2T from 2016 to 2018 using an improved retrieval algorithm. Firstly, the study
analyzes the retrieval errors of the MISR AOD for nine camera angles before the improved
retrieval. Secondly, we established an error correction model to correct the estimated surface
reflectance of MISR, thereby improving the surface reflectance at 9 angles. The improved surface
reflectance retrieval is used to obtain the MISR AOD with high accuracy. Finally, the improved
AOD retrieval method is verified and its estimated results are compared with the previous
retrieval.



## 2. MISR, MODIS, and AERONET Data

### 2.1 MISR and MODIS data



The MISR sensor is manufactured by National Aeronautics and Space Administration
(NASA) Jet Propulsion Laboratory (JPL) in the U.S. The MISR sensor consists of nine cameras,
each fixed at a specific angle of view along the orbital direction.The MISR has four bands (Blue:
446 nm, Green: 558 nm, Red: 672 nm and Near InfraRed: 866 nm) and nine angle (Table S1). The
MISR is capable of imaging the region almost simultaneously by all cameras within 7 min (Diner
et al., 1998; Martonchik et al., 2002; Kahn et al., 2007). 36 channels of MISR data are included,
all of which can be retrieved for AOD. Typically, medium angles are used for surface observations
and large angles are more sensitive to the effects of cloud cover and atmospheric aerosols. The
MISR product (MIL2ASAE_3) format is .nc, and the MISR format is HDF-EOS. The data used in
this study are MISR Level 1B2 Terrain Projected Data (MI1B2T) data. The projection of the
MI1B2T data is the Space Oblique Mercator (SOM), which uses Hierarchical Data Format - Earth
Observing System (HDF-EOS) strip record data, and out of 233 paths, each of the 233 orbits
consists of 180 mutually independent blocks (Kahn et al., 2005). The study extracted 64 and 65
blocks of data covering the Yangtze River Delta region (Fig. S1). MISR data cannot be directly
processed using Arcgis and ENVI due to its special storage method. The study uses HDF-EOS To
GeoTIFF Conversion Tool (HEGTool) for batch processing of MI1B2T and MI1B2GEOP. Using
the HEGTool, the radiation data in the MI1B2T dataset is extracted, and as the radiation data
contains 180 blocks, the corresponding blocks, output data types and projections are selected
based on the area locations. Solar zenith angle, solar azimuth angle, satellite zenith angle for 9
cameras and satellite azimuth angle for 9 cameras data were extracted from the MI1B2GEOP, and



the corresponding blocks, output data types and projection information were selected. The TIFF
data output by the software is only then available for the next step of processing using ARCGIS
and ENVI. For the downloaded MISR data, remote sensing images with cloud pixels less than
50% are used for cloud detection and removal of cloud pixels. This study using blue bands to set
thresholds to remove cloud pixels. After repeated experiments, if a fixed threshold is used, cloud
pixels cannot be removed cleanly from the 9 observation angles of MISR (Fig. S2, Fig. S3).
Therefore, this study uses dynamic threshold method to remove MISR cloud pixels from images.
The dataset used in the study is shown in table S2.

MODIS L1B data are Earth observation data stored in a hierarchical (HDF) format ,

providing MOD02QKM, MOD02HKM and MOD021KM data respectively (Hong et al., 2007;
Bandaru et al., 2013; Wong et al., 2020). MODIS geolocation data (MOD03/MYD03) contain the
MODIS solar/satellite zenith angle for each 1 km EV (Earth View) centre, the latitude, longitude,
solar/satellite azimuth and land/sea thresholds. To obtain the MODIS L1B apparent reflectance,
MOD03/MYD03 solar zenith angle data are also used. The MODIS L1B data used for the study
consisted of radiometric data (MOD02/MYD02) and geolocation data (MOD03/MYD03). The
MODIS L1B data were pre-processed with the corresponding MOD03 data for geometric
correction. The MODIS Conversion Toolkit (MCTK) was used to achieve radiometric calibration,
"bowtie" processing, geometric correction, reprojection and band extraction of MODIS data.
MODIS sensors can observe the surface at zenith angles up to approximately 65.5°, and repeated
observations of the same surface image over multiple days can be obtained for different angles
(-65.5° to 65.5°) of this image. Assuming that the surface of this image does not change
significantly during this time, this set of multi-angle observations can be used for the BRDF



(Bidirectional Reflectance Distribution Function) model retrieval. MODIS BRDF/Albedo is a
standard terrestrial level 3 product, generated from data acquired by the Terra and Aqua satellite
platforms and MODIS. This product has a 16-day cycle, with observations on day 9 of the 16-day
retrieval cycle being assigned a weight to obtain daily data, which is the global surface albedo
daily product data (Hsu et al., 2004). The core dataset of the MODIS BRDF product is
MCD43A1.
**2.2 AERONET data**
AERONET uses a French produced solar radiometer CE-318 instrument to obtain solar direct
spectral radiation measurements at 340nm, 380nm, 440nm, 500nm, 670nm, 870nm, 936nm,
1020nm, and 1640nm channels every 3 minutes at a field of view angle of 1.5°. The total
atmospheric water vapour content can be obtained from the 936 nm channel measurements and the
AOD values can be retrieval using the remaining channel measurements with an retrieval error of
about 0.01-0.02. Therefore, it can provide aerosol characterization parameters with high accuracy
and validate the aerosol parameters from satellite retrievals (Lu et al., 2019). AERONET has more
than 600 observing sites globally distributed over land and ocean, using a sun photometer as the
basic observing instrument, and most of the sites achieve daily data acquisition and unified
transmission to the network for centralized processing. It plays an important role in studying
global aerosol radiation effects, aerosol transport, validating radiative transfer models and
verifying satellite remote sensing aerosol results. The study therefore examines the accuracy of the
satellite remote sensing retrieval of AOD using the AOD measured by AERONET as the true
value.
AERONET provides observations of AOD, retrieval products and precipitable water





distributed over a wide range of aerosol patterns worldwide. There are three quality levels of AOD
data. Level 1.0 (unscreened), Level 1.5 (cloud-screened and quality-controlled), and Level 2.0
(quality-assured). The study area is mainly the Yangtze River Delta region of China, where
AERONET has a large number of sites, but only the Taihu and Xuzhou-CUMT sites provide
continuous data, while the rest of the sites have been acquiring data for a relatively short period of
time.
The amount of available data for AERONET AOD Level 2 is relatively small. To ensure
sufficient ground site validation data, this study selected Level 1.5 data with a large and
continuous amount of current observation data to verify the satellite remote sensing AOD obtained
from retrieval (Dubovik et al., 2000; Li et al., 2009) . In terms of time, the AERONET AOD Level
1.5 data selected for this study correspond to the MISR data for the three years from 2016-2018,
respectively.
**3. Methodology**
**3.1 Problems in the previous surface reflectance estimation method**
Accurate estimation of the contribution of surface reflectance has been the focus and
difficulty in the process of remote sensing retrieval of AOD (Remer et al., 2009; Gupta et al.,
2016). Previous study used MODIS BRDF data and MODIS V5.2 Algorithm to determine the
MODIS surface reflectance (Chen et al., 2021), and after spectral conversion between the two
sensors, MODIS and MISR, the MISR surface reflectance was obtained for nine angles. A look-up
table was constructed using the Second Simulation of Satellite Signal in the Solar Spectrum (6S),
and then the MISR surface reflectance of the 9 angles was combined to obtain the MISR AOD of





different angles by retrieval. This study found the variation pattern of the MISR AOD of the 9
angles. However, the error of the 9-angle retrieved MISR AOD was relatively large compared to
the AERONET AOD (Table S3).

In 6S model, a series of parameters related to the simulated imaging date atmospheric

conditions need to be input, including geometric parameters, AOD, water vapor, ozone, elevation,
etc. At the same time, these parameters are inputted through a simple and intuitive lookup table.
Finally, a linear atmospheric correction formula is generated to obtain the surface reflectance
values after 6S atmospheric correction for each pixel in each band one by one. The 6S model
inputs the geometric parameter information corresponding to the MISR image at the the Taihu and
Xuzhou-CUMT sites, and the input of 550nm AOD parameters is the AERONET AOD of the two
sites. Then, the 6S model is used to obtain the MISR atmospheric corrected reflectance. To
analyze the overall high AOD values retrieved from 9 angles of MISR, this study compared the
MISR atmospheric corrected reflectance at that pixel position with the MISR surface reflectance
(Fig. 1) (The calculation method for MISR surface reflectance refers to chen et al. (2021)). It can
be seen that the MISR surface reflectance is low compared to the value of MISR atmospheric
corrected reflectance at the corresponding locations of the two sites, resulting in a higher retrieve
MISR AOD compared to the AERONET AOD. Therefore, it is necessary to establish a correction
model to correct the MISR surface reflectance to improve the retrieval accuracy of the MISR
AOD.

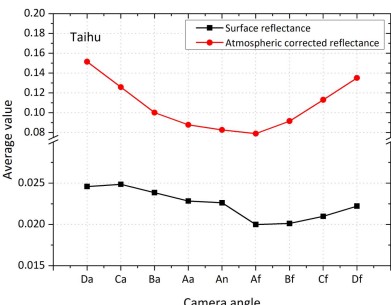
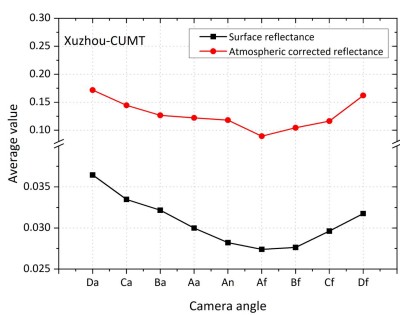

Figure 1. Comparison of MISR surface reflectance with atmospheric corrected reflectance in the

blue band (At the pixel locations of Taihu and Xuzhou-CUMT sites).

**3.2 Improved surface reflectance estimation method**

In order to develop a correction model to improve the surface reflectance, the design scheme

of this study is shown below:

a) Atmospheric correction of MODIS L1B using the 6S model to obtain MODIS atmospheric

corrected reflectance;

b) The new estimated MISR surface reflectance based on the MODIS atmospheric correction

was calculated by bringing the MODIS atmospheric correction reflectance into Eq. 1 and Eq. 2.
The MISR surface reflectance was combined with the newly estimated MISR surface reflectance
and a regression was fitted (with 60% of the overall sample data randomly selected) to create a
surface reflectance error correction model, as shown in the following formula:

$$\rho(\theta_s,\theta_v,\phi)_{MISR\_a} = \rho(\theta_s,\theta_v,\phi)_{MODIS\_at} \times \frac{BRDF(\theta_s,\theta_v,\phi)_{MISR}}{BRDF(\theta_s,\theta_v,\phi)_{MODIS}} \qquad (1)$$

In Eq. (1), $BRDF(\theta_s,\theta_v,\phi)_{MISR}$, $BRDF(\theta_s,\theta_v,\phi)_{MODIS}$ are BRDFs obtained at MISR and

MODIS angles, respectively. $\theta_s$ is the solar zenith angle, $\theta_v$ is the satellite zenith angle, and $\phi$ is
the relative azimuth angle. $\rho(\theta_s,\theta_v,\phi)_{MISR\_a}$ is the surface reflectance of MODIS at the



geometric observation angle of MISR, and $\rho(\theta_s,\theta_v,\phi)_{MODIS\_at}$ is the MODIS atmospheric
corrected reflectance.

This study selected spectral data containing 28 typical features of different types of

vegetation, soil and water bodies from five standard spectral libraries that come with the ENVI
software. Calculate the surface reflectance of different features in the blue bands of MODIS and
MISR using formulas (Chen et al., 2021).

$$\rho(\theta_s,\theta_v,\phi)_{MISR} = \rho(\theta_s,\theta_v,\phi)_{MISR\_a} \times 0.9834 - 0.0081 \tag{2}$$

The New MODIS surface reflectance ( $\rho(\theta_s,\theta_v,\phi)_{MISR\_a}$ ) at the MISR angle obtained from

Eq. (1) is converted to the MISR surface reflectance by Eq. (2).

c) The MISR surface reflectance estimated by Eq. 2 is transformed by an error correction

model to obtain the final improved MISR surface reflectance. The improved MISR surface
reflectance will be used in the retrieval of the AOD. The MISR correction model was developed
by fitting a linear regression of the previously estimated MISR surface reflectance based on the
MODIS V5.2 algorithm to the MISR surface reflectance estimated based on the MODIS
atmospheric correction (60% of the data were randomly selected) as shown in Eq. 3. The
previously estimated 9-angle MISR surface reflectance was error-corrected by Eq. 3 to obtain the
improved surface reflectance for the 9 angles of the MISR sensor, which was ultimately used to
perform the MISR AOD retrieval for the 9 angles.

$$\rho(\theta_s,\theta_v,\phi)_{MISR-b}^{*} = \rho(\theta_s,\theta_v,\phi)_{MISR} \times 0.9209 + 0.0409 \tag{3}$$

Where $\rho_{MISR-b}^{*}$ is the improved MISR surface reflectance in Eq. (3).



### 3.3 Flow of improved multi-angle AOD retrieval


The flow of the improved surface reflectance algorithm for this study is shown in Fig. 2. The
MODISL1B data were first atmospherically corrected using 6S, and then the MISR surface
reflectance estimated from previous study was combined with the new MISR surface reflectance
estimated from Eq. 2 to build a MISR error correction model to obtain the improved MISR surface
reflectance (Chen et al., 2021). The study retrieved the MISR AOD for nine camera angles using
improved MISR surface reflectance. We use AERONET AOD to validate the improved MISR
AOD. Compare the improved AOD with the previously retrieved AOD, and analyze the accuracy
and spatial distribution trend of the improved AOD. The AOD retrieval method used in this study
is based on chen et al. (2021). The selection of appropriate aerosol type is very critical for the
retrieval of aerosol optical depth. It has been shown that continental aerosols can be used for AOD
retrieval in the Yangtze River Delta (He et al., 2015). Therefore, we used continental aerosols in
our aerosol retrieval for the study area.



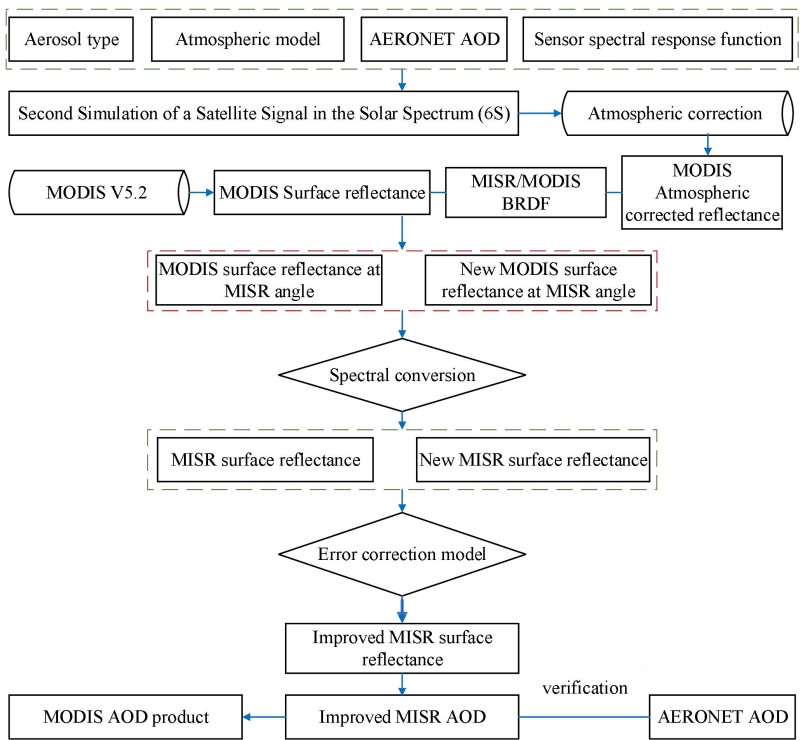


Figure 2. Flow chart of the improved MISR surface reflectance algorithm

## 4. Results and discussion

### 4.1 Improved MISR surface reflectance variation characteristics

The estimated MISR surface reflectance, the MISR atmospheric corrected reflectance and the

improved MISR surface reflectance are presented in the Fig. 3. This is the average of all sample
data at the corresponding locations at the two sites in Taihu and Xuzhou-CUMT for the valid dates
of 2016-2018. It can be noted that at the two site of Taihu and Xuzhou-CUMT, the 9 camera angle
MISR-improved surface reflectance values are overall higher than the MISR surface reflectance
and lower than the MISR atmospheric corrected reflectance. The nine camera angle MISR surface
reflectance values ranged from 0.02 to 0.04. The improved surface reflectance averages were
overall greater than the previously estimated MISR surface reflectance.

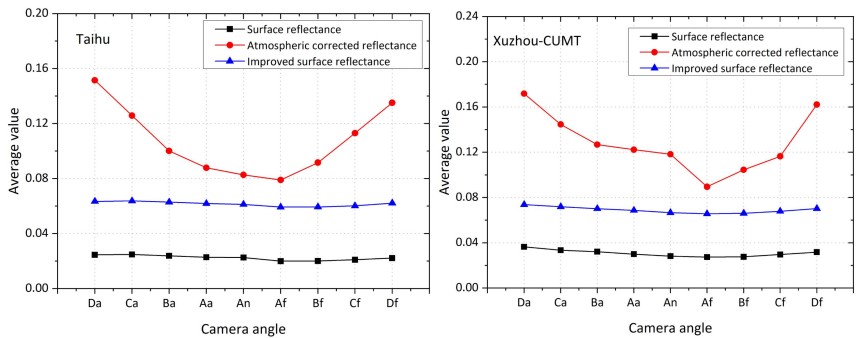

**Figure 3.** Comparison of MISR surface reflectance, atmospheric corrected reflectance and
improved surface reflectance in the blue band (This is the multi-year average of the sample data
for the two sites in Taihu and Xuzhou-CUMT).
To clarify the trend of the improved surface reflectance, the study performed a time-series
analysis of the MISR surface reflectance and the improved surface reflectance (Fig. 4). It can be
seen that the improved MISR surface reflectances are all higher than the previously estimated
MISR surface reflectances. MISR surface reflectance values generally range from 0-0.05, with
improved surface reflectance values ranging from approximately 0.05-0.1. The improved surface
reflectance values have increased overall.

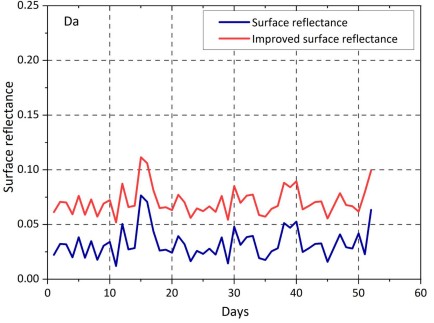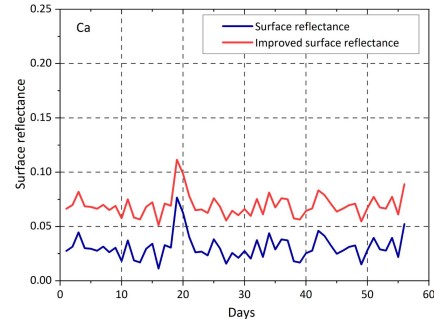

**Figure 4.** Surface reflectance time series of MISR sensors in the blue band at 9 observation angles.

(The order of time from front to back for 9 angles is shown in Table S2)





## 4.2 Results of the Improved MISR AOD retrieval


MISR AOD for 2016-2018 were obtained using an improved surface reflectance retrieval.
The study presents retrieval results from nine camera observation angles of the MISR sensor on 12
June 2018 (Fig. 5). As can be seen from the spatial distribution of AOD, the retrieval results in the
study area do not exceed a value of 1. The overall spatial distribution trend is generally consistent
with the results before the improvement (Chen et al., 2021), but differs in the magnitude of the
values. Values in the north-eastern and southern regions range from 0.5 to 1, indicating to some
extent that the air quality in this region is poor. The five camera observation angles, Ba, Aa, An,
Af and Bf, retrieve AOD values in the approximate range of 0.25-0.5. In the central region, the
four camera observation angle values of Da, Ca, Cf and Df were mostly in the range of 0-0.25.
The values indicate that the air quality in the region is generally good, but there are some areas of
light air pollution. The higher AOD in the southern part of Shandong Province and the northern
part of Jiangsu Province may be due to increased local aerosol emissions caused by human
activities.

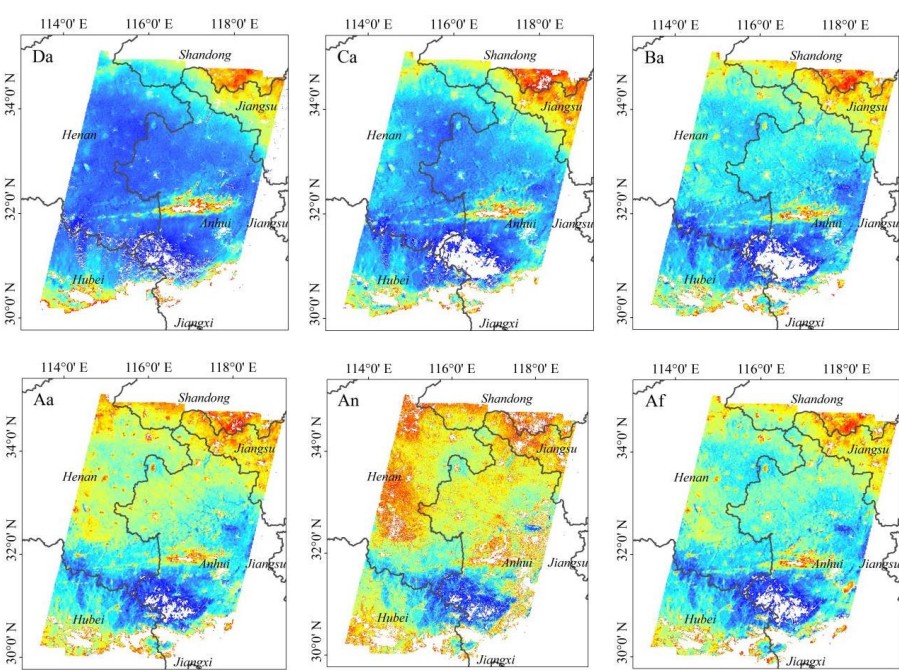



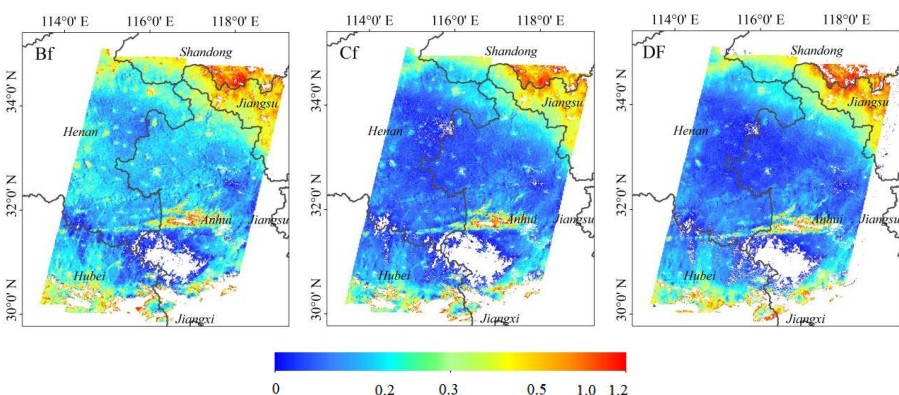

**Figure 5.** Plot of AOD 550nm retrieval results for the improved MISR 9 camera angles on 12 June 2018.

The study validates our improved MISR AOD spatial distribution results by comparing with MODIS AOD products of the same date (Fig. 6). The MODIS AOD products have a resolution of 3km. It can be seen that the trend of spatial distribution of MODIS AOD products is consistent with the improved MISR AOD. However, the MODIS AOD product has more missing data, which can be avoided by the AOD obtained from the retrieval of the improved algorithm, and the AOD retrieval by the improved algorithm has a higher resolution by comparing with the image quality of the MISR AOD product.

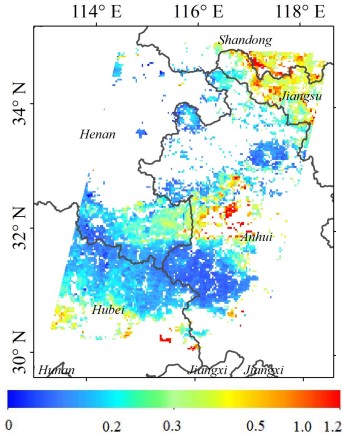

**Figure 6.** MODIS AOD 550nm product spatial distribution on June 12, 2018



### 4.3 Verification of the improved MISR AOD


There are many AERONET sites in the Yangtze River Delta region, but so far, only the
Taihu and Xuzhou-CUMT sites continue to provide data, and other sites have a short time to
obtain data. Therefore, the Taihu and Xuzhou-CUMT sites with more data are selected for
verification. To verify the retrieved MISR AOD, In terms of time, we selected effective AOD
records in the 550 nm band within a 30 minute interval between the AERONET ground
observation site and the Terra satellite. The 9 camera views of MISR require about 7 minutes to
observe the same geographical location, with relatively short intervals. Therefore, we will use the
calculated AERONET AOD average as the approximate truth value, and compare the average
value with the retrieved MISR AOD to verify and reduce errors caused by time difference. In
terms of space, we selected pixels observed by MISR sensors from 9 angles and compared them
with the nearest data observed by AERONET, which can reduce errors caused by spatial
differences. The solar photometer does not have a 550 nm wavelength that corresponds to the
retrieval results, and the AOD at 550 nm can be calculated by applying Angstrom (Eq. 4).

$$\tau(\lambda) = \beta\lambda^{-\alpha} \tag{4}$$

In the formula, $\tau(\lambda)$ is the AOD at wavelength $\lambda$, $\beta$ is the concentration of the entire
atmospheric aerosol, and $\alpha$ is the wavelength index of Angstrom.
In this study, four parameters will be used to assess the accuracy of the remotely sensed AOD
dataset, namely the correlation coefficient (R), the Root Mean Square Error (RMSE), p-value and
the relative mean bias (RMB). The specific calculation principles for the three parameters R,
RMSE and RMB are shown in Eq. (5)-(7). The validation results of this study's improved AOD
dataset from 2016-2018 at Taihu and Xuzhou-CUMT sites are shown in Fig. 7 and Fig. 8.
In general, the scatter plot is distributed above and below the 1:1 line. R is a parameter used
to characterize the correlation between the remote sensing retrieval results and the ground-based
retrieval results. At the Taihu site, R reach up to 0.89. At the Xuzhou-CUMT site, R reach up to
0.85. The RMSE is a parameter used to characterise the absolute error of the remote sensing
retrieval results, with a minimum root mean square error of 0.21 at the Taihu site and 0.20 at the



Xuzhou-CUMT site. RMB is the parameter used to characterise the relative error of the remote
sensing retrieval results, with a minimum RMB of 0.52 at the Taihu site and 0.32 at the
Xuzhou-CUMT site. In summary, by comparing the results with the validation of the AOD scatter
plot before the improvement, the accuracy of the nine camera observation angles at both sites has
improved after the improvement (Table 1).
$$R = \frac{\sum_{i=1}^{N}(A_i - \overline{A})(A_i' - \overline{A'})}{\sqrt{\sum_{i=1}^{N}(A_i - \overline{A})^2 \sum_{i=1}^{N}(A_i' - \overline{A'})^2}} \quad (5)$$

$$RMSE = \sqrt{\sum_{i=1}^{N}(A_i - A_i')^2 / N} \quad (6)$$

$$RMB = \sum_{i=1}^{N}(A_i - A_i') / N \quad (7)$$

where $A_i$ is the retrieve MISR AOD, $A_i'$ is the corresponding AERONET AOD, $\overline{A}$
and $\overline{A'}$ are the mean values of the retrieve MISR AOD and AERONET AOD, respectively. $N$ is
the number of valid matching results for AERONET AOD and MISR AOD.

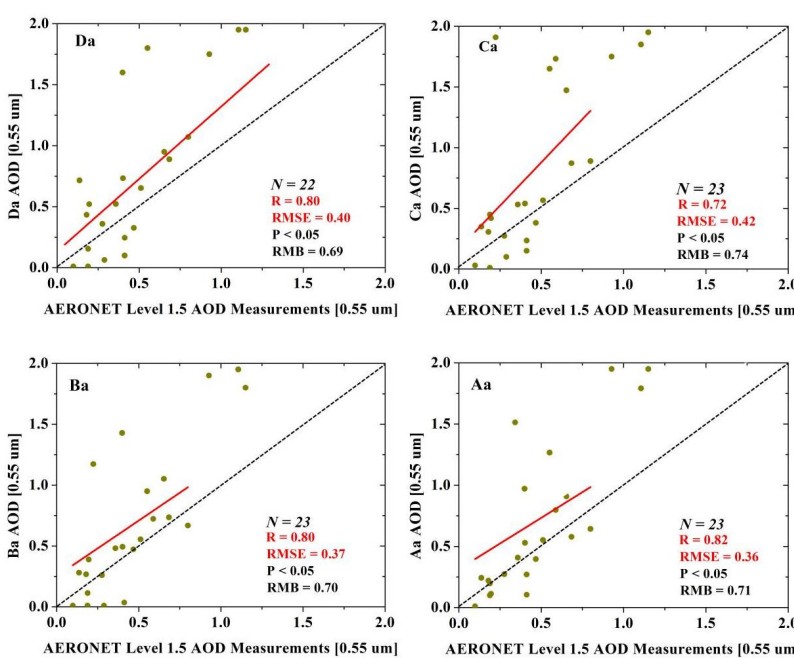



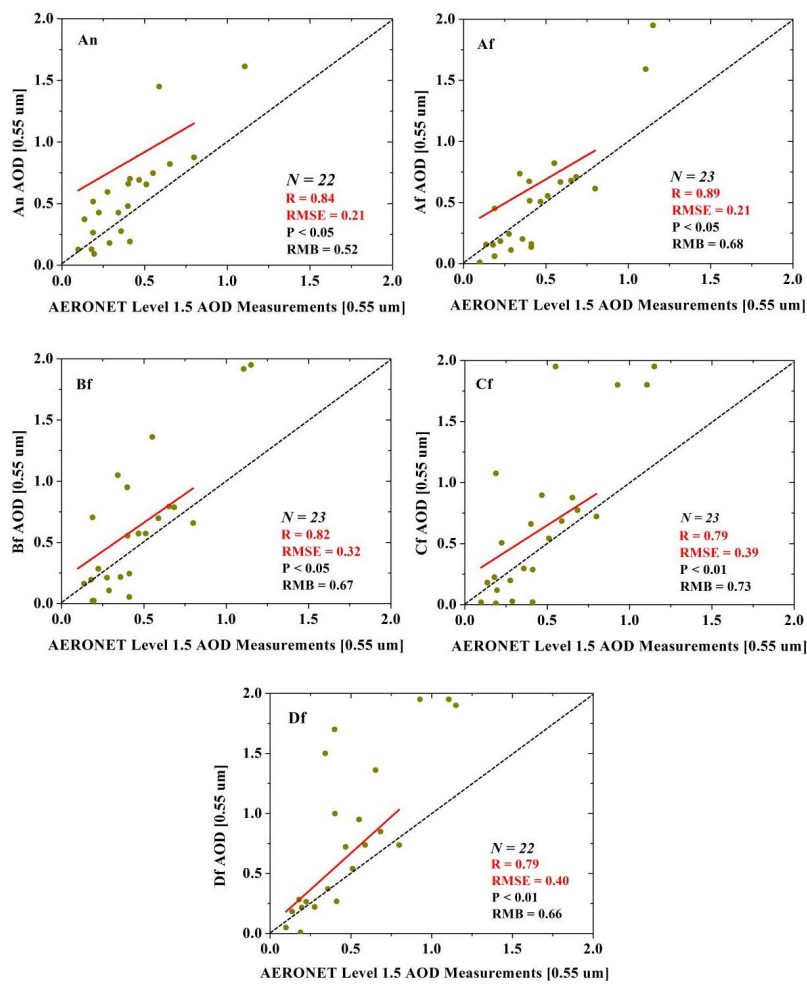

**Figure 7.** Comparison between improved MISR AOD and AERONET AOD at Taihu site (N is

the number of verification points, red line represents a linear fitting line).

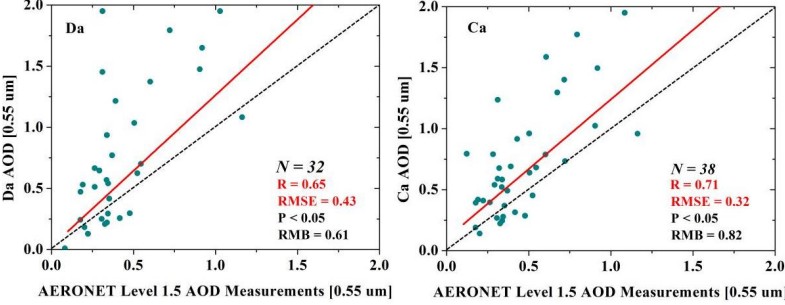



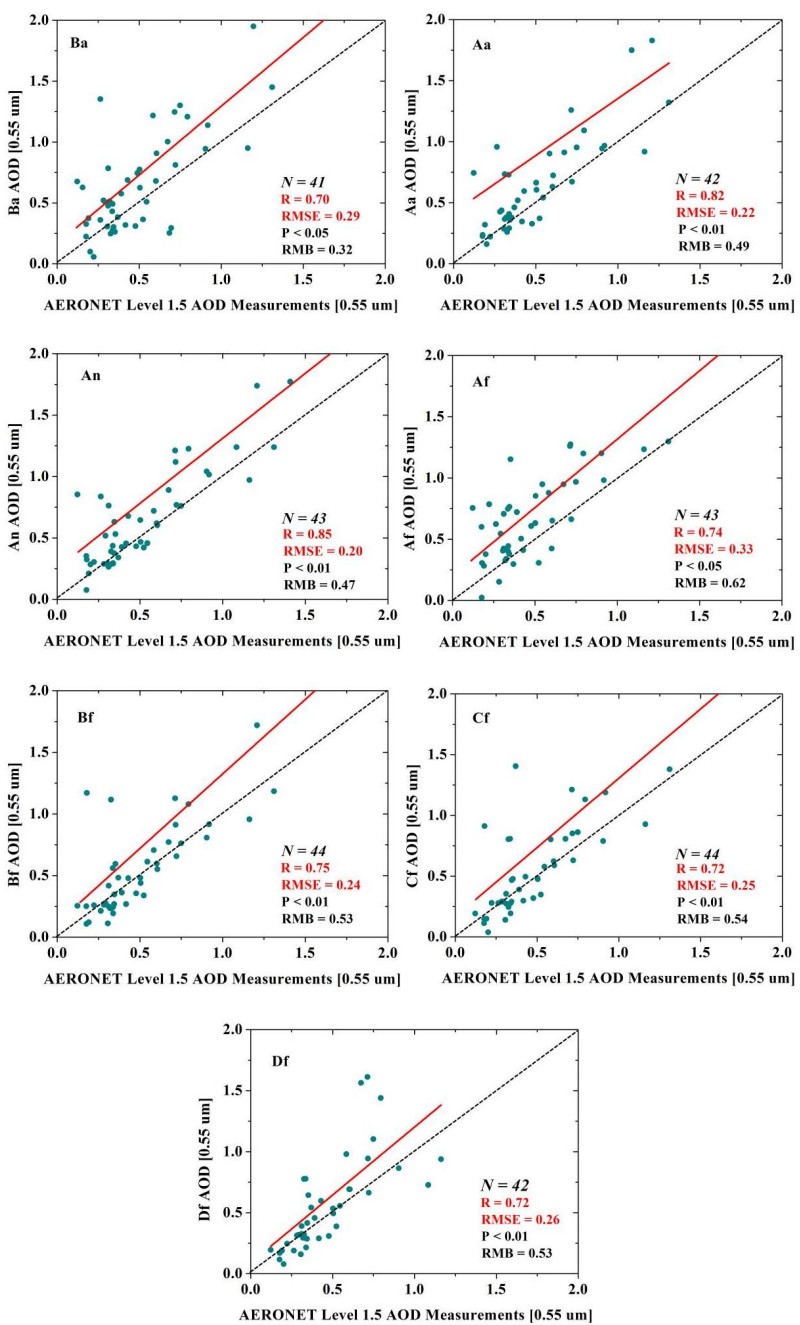

**Figure 8.** Comparison between improved MISR AOD and AERONET AOD at Xuzhou-CUMT
site (N is the number of verification points, red line represents a linear fitting line).







**Figure 9.** Comparison of validation of retrieved AOD with MODIS AOD product

**Table 1.** Precision comparison of MISR AOD and AERONET AOD before

and after improvement

| Site | Angle | R | RMSE | RMB | Improved R | Improved RMSE | Improved RMB |
|------|-------|------|------|------|------------|---------------|--------------|
| Taihu | Da | 0.77 | 0.20 | 1.08 | 0.80 | 0.40 | 0.69 |
| | Ca | 0.70 | 0.29 | 1.00 | 0.72 | 0.42 | 0.74 |
| | Ba | 0.77 | 0.14 | 0.61 | 0.80 | 0.37 | 0.70 |
| | Aa | 0.81 | 0.11 | 0.68 | 0.82 | 0.36 | 0.71 |
| | An | 0.45 | 0.29 | 1.22 | 0.84 | 0.21 | 0.52 |
| | Af | 0.72 | 0.14 | 0.87 | 0.89 | 0.21 | 0.68 |
| | Bf | 0.72 | 0.17 | 0.60 | 0.82 | 0.32 | 0.67 |
| | Cf | 0.57 | 0.24 | 0.65 | 0.79 | 0.39 | 0.73 |
| | Df | 0.77 | 0.20 | 0.47 | 0.79 | 0.40 | 0.66 |
| Xuzhou-CUMT | Da | 0.45 | 0.36 | 1.58 | 0.65 | 0.43 | 0.61 |
| | Ca | 0.59 | 0.34 | 0.96 | 0.71 | 0.32 | 0.82 |
| | Ba | 0.67 | 0.27 | 0.78 | 0.70 | 0.29 | 0.32 |
| | Aa | 0.73 | 0.24 | 0.78 | 0.82 | 0.22 | 0.49 |
| | An | 0.75 | 0.20 | 0.85 | 0.85 | 0.20 | 0.47 |
| | Af | 0.72 | 0.23 | 0.63 | 0.74 | 0.33 | 0.62 |
| | Bf | 0.62 | 0.28 | 0.65 | 0.75 | 0.24 | 0.53 |
| | Cf | 0.68 | 0.28 | 0.66 | 0.72 | 0.25 | 0.54 |
| | Df | 0.67 | 0.30 | 0.65 | 0.72 | 0.26 | 0.53 |

By comparing the validation results of MODIS AOD products with those of observation sites

(Taihu: R=0.59, RMSE=0.19, P<0.05, RMB=0.52; Xuzhou-CUMT: R=0.71, RMSE=0.25, P<0.05,

RMB=0.44) (Chen et al., 2021), we find that the improved MISR AOD has a higher correlation





with MODIS AOD products in the Taihu and Xuzhou-CUMT sites. The smaller the observation
angle of the improved MISR AOD, the closer the error is to that of the MODIS AOD product. The
observation angle of MISR An is the same as that of MODIS. Therefore, we selected An
observation angle and MODIS AOD products at two pixel positions in the Taihu and
Xuzhou-CUMT for verification (Fig. 9). The results show that the An AOD retrieval by the
improved algorithm correlates well with the MODIS AOD product, and the position errors of the
two image elements are close to each other. The RMSE of Xuzhou-CUMT site is slightly higher
than that of Taihu site, and the RMB of Taihu site is slightly higher than that of Xuzhou-CUMT
site.
**5. Conclusion**

This study first explored the problem of estimating the surface reflectance in our previous

study and then obtained an error correction model for the surface reflectance using a linear fit of
the MISR surface reflectance and a new estimate of the MISR surface reflectance. The improved
MISR surface reflectance was obtained by means of an error correction model. A new AOD
product was retrieved using the improved surface reflectance and a look-up table constructed from
6S model. Two AERONET ground observation sites with longer time series were used to validate
the AOD obtained by satellites.

(1) Overall, the improved AOD and its spatial distribution trends are consistent with our

previous results. The AOD estimated by our improved method presented a higher accuracy and a
high degree of agreement with the AERONET ground-based observational AOD.

(2) More importantly, compared to the MODIS AOD products, the retrieved AOD in this

study has fewer missing AOD pixels and finer spatial resolution. The retrievals of An Angle AOD
by the improved algorithm are highly correlated with the MODIS AOD products, as shown
through validation with the MODIS AOD product.



(3) In the future, more aerosol models conforming to the actual situation in the study area
can be constructed using the AERONET ground observation data and introduced into the MISR
AOD retrieval algorithm to further improve the accuracy of the AOD retrieval results. In this study,
the AERONET AOD was used as the true value and as an input to the AOD parameter in the 6S
model for atmospheric correction of MISR and MODIS images. A surface reflectivity error
correction model was then obtained to retrieve the AOD for the entire region. It should be
emphasized that the more AERONET sites used to train the corrected model, the more accurate
the AOD results obtained by this method. However, the data of AERONET ground observation
sites was limited. In the future, the study area can be expanded to large scale and longer time
series.
**Acknowledgement**

This study was supported by the Key Program of the National Natural Science Foundation of

China (42130609), Jiangsu Funding Program for Excellent Postdoctoral Talent (2023ZB482), the
Natural Science Foundation of Jiangsu Province of China (BK20220455), and the National
Science Foundation of China (42201028).
**Competing interests**

The contact author has declared that none of the authors has any competing interests.



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
