# Peer review of "Multi-angle aerosol optical depth retrieval method based on"

_Atmospheric Measurement Techniques, 2023_

## Author Comment (AC1)

**Response to the reviewers' comments**

We are very grateful to the editor and reviewer for providing constructive comments on our manuscript and for giving us the opportunity to revise the manuscript. We have carefully considered the reviewers' comments and revised the manuscript accordingly. The responses to the comments or suggestions are shown below. Comments are shown in black font and **our responses are shown in blue font.**

**Reviewer(s) Comments:**

**Reviewer #3**

**General comments:**

The authors improve upon their own prior work to estimate the surface reflectance from MISR measurements, and then use the improved estimate in conjunction with a look-up table-based approach to obtain a new aerosol optical depth product. They validate their product with AERONET measurements. They contend that the new product agrees well with AERONET data and has a higher spatial resolution and fewer missing pixels than the MODIS product.

There are a number of major flaws in the manuscript. The new methodology is not explained clearly. The description is convoluted and the novelty is not obvious. While the new product does have less missing pixels than the MODIS product, the improvement compared to existing products seems to be marginal. In my opinion, the authors need to do a significant amount of work to make this suitable for publication in AMT. For this reason, I would recommend rejection of the manuscript as currently written.

**Response:** We thank the reviewer for providing valuable and thought-provoking comments. We apologize for the poor language of our manuscript. Actually, we worked on the manuscript for a

long time and the repeated addition and removal of sentences and sections obviously led to poor readability. We have now worked on both language and readability. We really hope that the flow and language level have been substantially improved. In addition, we have revised and improved the manuscript where it was not clearly presented.

**Main comments:**

1. Sections 2.1 and 2.2: There is information overload here. The authorize should summarize in one or two paragraphs exactly what information is used and from what datasets.

**Response:** We have recognized the issue of information overload in Sections 2.1 and 2.2. In order to enhance readability and clarity, we have made revisions. We have accurately summarized the information used and the sources of the datasets in one or two paragraphs. (Section 2 )

2. Section 3.1: The description here is very confusing and hard to follow.

**Response:** We deeply apologize for any inconvenience caused by this. We have made logical adjustments and simplifications to the sentences in Section 3.1 to facilitate better understanding and readability for the reviewers. The specific modifications are as follows:

" The accurate estimation of surface reflectance is a crucial and challenging aspect in the retrieval of AOD from satellite remote sensing data (Remer et al., 2009; Gupta et al., 2016). Previous research has identified the variation patterns of 9-angle MISR AOD (Chen et al., 2021). However, the AOD retrieved at 9 angles exhibits relatively large errors when compared to AERONET AOD (Table S3). Atmospheric correction can eliminate the effects of clouds and aerosols on data, obtaining the true surface reflectance. When using the 6S model to calculate atmospheric correction reflectance for the MISR sensor, several parameters need to be inputted,

including geometric parameters, AOD, water vapor, ozone, altitude, and radiation data, etc. In this study, we inputted the MISR geometric parameters and radiation data corresponding to Taihu and Xuzhou-CUMT stations, while the AOD parameter inputted was the AERONET AOD for these two stations. Through the generated linear atmospheric correction formula, we calculated the atmospheric correction reflectance for each pixel. To investigate the reasons for the higher AOD values retrieved from the 9 MISR angles, this study compared the MISR atmospheric correction reflectance and MISR surface reflectance at the pixel location (Figure 1) (MISR surface reflectance calculation method referenced Chen et al. (2021)). It was observed that the MISR surface reflectance was relatively lower compared to the MISR atmospheric correction reflectance. As a result, the retrieved MISR AOD values were higher compared to AERONET AOD. Therefore, it is necessary to establish a correction model to adjust the MISR surface reflectance and improve the retrieval accuracy of MISR AOD. ”

3. Section 3.2: On the other hand, there needs to be more detail here about how exactly the surface reflection is corrected to account for atmospheric effects as that is critical to the topic addressed by the authors. How exactly is the 6S model used for the atmospheric correction step a? More details need to be provided here, such as how atmospheric parameters relevant to the RT calculation (gas absorption? Aerosol optical depth? Aerosol layer height? Surface BRDF model?) are chosen.

**Response:** We apologize for not providing detailed explanations regarding the atmospheric correction process. In our manuscript, we have included an explanation of how the 6S model is

used to perform atmospheric correction on MODIS radiance data. The following is a detailed explanation:

"When using the 6S model to calculate the atmospheric correction reflectance of the MODIS sensor, several parameters need to be provided. These parameters include geometric parameters, AOD, atmospheric models, aerosol types, sensor radiance data, sensor altitude, and spectral parameters. In this study, we inputted the MODIS geometric parameters and radiance data corresponding to the Taihu and Xuzhou-CUMT sites. The AOD parameter was obtained from the AERONET AOD data measured at these two sites. The atmospheric models chosen were mid-latitude winter and mid-latitude summer to account for seasonal variations in atmospheric transmission. The aerosol type selected was continental aerosol since it is typically found in the Yangtze River Delta region. The sensor altitude was set to the height of satellite observations. The spectral parameters were defined based on the wavelength bands of the MODIS sensor. By providing these parameters, we can utilize the 6S model to calculate the atmospheric correction reflectance of the MODIS sensor."

4. In Equation (1), how are the MISR and MODIS BRDF obtained?

**Response:** We apologize for not providing a description of how MISR and MODIS BRDF were obtained. Below is a detailed explanation:

"We utilized the MODIS BRDF/Albedo product MCD43A1 data and employed the Ross-Li model to simulate surface bidirectional reflectance under MODIS and MISR observation geometries. The linear kernel-driven BRDF model comprises three essential parameters: the

nadir-view reflectance, and the weighting coefficients for the two kernel functions. The model can

be expressed using the following formula:

$$BRDF(\theta_s, \theta_v, \phi) = f_{iso}(\Lambda) + f_{vol}(\Lambda)K_{vol}(\theta_s, \theta_v, \phi) + f_{geo}(\Lambda)K_{geo}(\theta_s, \theta_v, \phi) \qquad (1)$$

$$K_{vol}(\theta_s, \theta_v, \phi) = \frac{(\pi/2 - \xi)\cos\xi + \sin\xi}{\cos\theta_s + \cos\theta_v} - \frac{\pi}{4} \qquad (2)$$

$$K_{geo}(\theta_s, \theta_v, \phi) = O(\theta_s, \theta_v, \phi) - \sec\theta'_s - \sec\theta'_v + \frac{1}{2}(1 + \cos\xi')\sec\theta'_s \sec\theta'_v \qquad (3)$$

$$O(\theta_s, \theta_v, \phi) = \frac{1}{\pi}(t - \sin t \cos t)(\sec\theta'_s + \sec\theta'_v) \qquad (4)$$

$$\cos t = \frac{h}{b}\frac{\sqrt{D^2 + (\tan\theta'_s \tan\theta'_v \sin\phi)^2}}{\sec\theta'_s + \sec\theta'_v} \qquad (5)$$

$$D = \sqrt{\tan^2\theta'_s + \tan^2\theta'_v - 2\tan\theta'_s \tan\theta'_v \cos\phi} \qquad (6)$$

$$\cos\xi' = \cos\theta'_s \cos\theta'_v + \sin\theta'_s \sin\theta'_v \cos\phi \qquad (7)$$

$$\theta'_s = \tan^{-1}(\frac{b}{r}\tan\theta_s) \qquad (8)$$

$$\theta'_v = \tan^{-1}(\frac{b}{r}\tan\theta_v) \qquad (9)$$

In the aforementioned equation, $BRDF(\theta_s, \theta_v, \phi)$ represents the bidirectional reflectance of

the surface, while $\theta_s$, $\theta_v$, and $\phi$ denote the solar zenith angle, view zenith angle, and relative

azimuth angle, respectively. $\Lambda$ stands for the bandwidth, while $K_{vol}(\theta_s, \theta_v, \phi)$ and

$K_{geo}(\theta_s, \theta_v, \phi)$ represent the volumetric scattering kernel and geometric optical scattering kernel,

respectively. These terms are all functions of the incident and viewing angles. $f_{iso}$, $f_{vol}$, and

$f_{geo}$ correspond to the weights assigned to isotropic scattering, volumetric scattering, and

geometric optical scattering in the reflectance, which serve as coefficients for their respective kernel functions. $\xi$ represents the scattering angle, while b, h, and r represent the vertical radius, horizontal radius, and height of the sphere's center, respectively. Based on empirical values, these three parameters can be considered as fixed values. Within the production process of MODIS BRDF model parameter products, the following relationships exist among these parameters: h/b=2 and b/r=1 (Schaaf et al., 1999). By utilizing extrapolation with kernel functions, the surface's bidirectional reflectance can be computed under arbitrary solar incident and satellite viewing directions using the aforementioned equation. "

5. Conceptually, what is the difference between the "MODIS surface reflectance at the MISR angle" and the "MISR surface reflectance" ?

**Response:** The MODIS surface reflectance at the MISR angle refers to the MODIS surface reflectance calculated based on the MISR observation angles. On the other hand, the MISR surface reflectance refers to the estimated reflectance of the MISR sensor itself.

6. Section 4.1: How do we know that the "improved" surface reflectance is more reflective of reality?

**Response:** The objective of atmospheric correction is to mitigate the influence of factors like clouds and aerosols on the data, thereby obtaining the true surface reflectance. The ultimate outcome of this process is the surface reflectance. Consequently, we compared the refined MISR surface reflectance with the reflectance obtained through atmospheric correction.

7. Figure 5 should be accompanied by a figure showing the results using the old surface reflectance data, so that the changes in the AOD results can be clearly seen.

**Response:** We would like to thank the reviewer for his/her valuable feedback. To prevent duplication, we have omitted the results of AOD obtained through the previous study's application of the older surface reflectance retrieval technique in our manuscript. These findings are available in the work by Chen et al. in 2021.

8. Figure 7: There is a lot of scatter in the results. I am not sure the new method is doing as good a job as the authors claim.

Figure 8: These results are better but there is still a fair bit of systematic bias compared to AERONET data.

**Response:** Based on Table 1, it is evident that the results from Figures 7 and 8 exhibit higher correlation and overall smaller relative deviations compared to the previous old retrieval results. The improved retrieval results have been validated against AERONET AOD, showing significant improvement in terms of correlation and deviation compared to the old retrieval results validated against AERONET AOD. Although there are still some systematic biases present, it is clear that the retrieval results have been improved.

9. Again, the old results should be overplotted to show the improvement.

**Response:** Thank you very much for the reviewer's comments. We compared the old retrieval results with the improved retrieval results at the corresponding positions of two sites in Table 1.

We found that the improved retrieval results exhibit better correlation and overall smaller relative deviations compared to the old retrieval results.

10. Figure 9: The comparison with MODIS is also not very good.

**Response:** In our previous work, we validated MODIS AOD products against AERONET AOD using scatter plots (Fig. 1). By comparing the results from the following figure with the improved retrieval results presented in this manuscript, it is clear that the improved retrieval results have higher correlation and smaller deviation compared to the retrieval results from MODIS AOD products. Therefore, we believe that the improved AOD yields better retrieval performance than the MODIS AOD.

[Figure]

Fig. 1. MODIS AOD 550nm validation results for the corresponding locations at both sites. (Chen et al., 2021)

11. Table 1 is very confusing and there does not seem to be any explanatory description of the results therein.

**Response:** We apologize for not clearly explaining Table 1, and we have added an explanation of the contents to the manuscript. The details are as shown below:

"R represents the correlation between the old AOD retrieval results and AERONET AOD. RMB represents the relative deviation between the old algorithm-retrieved AOD and AERONET

AOD. Improved R represents the correlation between the improved AOD retrieval results and AERONET AOD. Improved RMB represents the relative deviation between the AOD retrieved using the improved algorithm and AERONET AOD."

**Technical comments:**

The paper is very poorly written and extremely hard to understand. There are grammatical and typo errors littered throughout the document. Here are some examples of awkwardly phrased sentences:

**Response:** We regret that there are grammatical and spelling errors as well as unclear expressions in the paper. We will carefully examine and correct these errors, and ensure that we pay more attention to grammar and spelling accuracy in future writing. Regarding the examples of awkward sentences you mentioned, we apologize for this. We will review these sentences and make modifications to improve their fluency and comprehensibility. We sincerely appreciate your valuable feedback, which is very helpful for us to improve the quality of the paper. We will closely consider your suggestions and take measures to ensure that future writing is more accurate, clear and easy to read.

12. Lines 115-116: 36 channels of MISR data are included, all of which can be retrieved for AOD.

**Response:** We have made language revisions to Section 2.1 and removed any unnecessary information to make the manuscript more accurate, clear, and readable.

13. Lines 164-165: Therefore, it can provide aerosol characterization parameters with high accuracy and validate the aerosol parameters from satellite retrievals

**Response:** We have revised the sentence as follows: "Therefore, AERONET provides high-precision aerosol characteristic parameters and can be used to validate satellite-retrieved AOD."

14. Lines 198-199: In 6S model, a series of parameters related to the simulated imaging date atmospheric conditions need to be input

**Response:** We have revised the sentence as follows: "When using the 6S model to calculate atmospheric correction reflectance for the MISR sensor, several parameters need to be inputted. "

15. Lines 257-258: Compare the improved AOD with the previously retrieved AOD, and analyze the accuracy and spatial distribution trend of the improved AOD.

**Response:** We have revised the sentence as follows: " We compared the improved AOD with the previously retrieved AOD and analyzed the accuracy and spatial distribution trends of the improved AOD. "

16. Line 124: Arcgis, ENVI not defined

**Response:** The full name of ArcGIS is "ArcGIS Geographic Information System" and the full name of ENVI is "Environment for Visualizing Images". These two software tools were used in our manuscript, and we have added relevant information about them in the corresponding sections.

17. Line 163: AOD values can be retrieval -> AOD values can be retrieved; an retrieval error -> a retrieval error

**Response:** We deeply apologize for the language errors found in the paper. We have carefully checked and corrected these errors.

---

## Author Comment (AC2)

**Response to the reviewers' comments**

We are very grateful to the editor and reviewer for providing constructive comments on our manuscript and for giving us the opportunity to revise the manuscript. We have carefully considered the reviewers' comments and revised the manuscript accordingly. The responses to the comments or suggestions are shown below. Comments are shown in black font and **our responses are shown in blue font.**

**Reviewer(s) Comments:**

**Reviewer #2**

**General comments:**

In this manuscript, mainly based on the first author's previous works (Chen et al., Tellus B: Chemical and Physical Meteorology, 2021, 73, 1940758; Chen et al., Advances in Space Research, 2021, 67, 858-867) and with the same MISR and AERONET datasets (MISR data with 9 camera angles on June 12, 2018 for AOD retrieval map show, Taihu and Xuzhou AERONET sites), an improved linear correction Equation (Eq. 3) is used to update the MISR surface reflectance. With the new updated MISR surface reflectance, the aerosol optical depths (AODs) of 9 camera angles are retrieved by the lookup table by 6S, respectively, and further systematically validated by the AERONET, as well as the MODIS AOD products.

**Response:** We thank the reviewer for his/her positive suggestions and valuable comments. Point-by-point responses to the reviewer are shown below.

**Specific Comments:**

1. However, this manuscript didn't explain in what way to extract the semi-empirical relationship in Eq. (3), which plays a most import role in the completeness and logic of this manuscript. Besides, since the title of manuscript is about "multi-angle aerosol optical depth retrieval method", it is a great pity that this manuscript didn't discuss how to retrieve the AODs and other key aerosol optical parameters by taking the full advantage of 9 camera angles' measurements together with the Eq. (3), which is also very important to improve the MISR's aerosol retrieval accuracy. Moreover, the structure and content of this paper are very similar to previously published paper (Chen et al., Tellus B: Chemical and Physical Meteorology).

**Response:** We apologize for not clearly describing the details of the algorithm in this manuscript. The following are explanations and supplements to the content of the manuscript.

① According to Equation 12, the MISR calibration model is established by performing a linear regression fit between the previously estimated MISR surface reflectance based on the MODIS V5.2 algorithm and the newly estimated MISR surface reflectance based on MODIS atmospheric correction (randomly selecting 60% of the data).

② Applying Equation 12, the previously estimated surface reflectance of the MISR sensor at 9 angles is corrected for errors, resulting in improved surface reflectance values. These improved surface reflectance values are then used to retrieve the MISR Aerosol Optical Depth (AOD) at the same 9 angles.

After carefully considering the reviewer's comment, we have added to the manuscript what is not clearly explained. The manuscript analyzes the reasons for the overall high MISR AOD values previously retrieved by Chen et al. (2021), mainly due to the underestimation of surface reflectance. To obtain more accurate surface reflectance, the manuscript focuses on exploring

improvements to the MISR surface reflectance algorithm. The final improved surface reflectance was utilized to obtain a more accurate MISR AOD dataset. This manuscript is a follow-up exploration and algorithmic improvement based on previous research. Therefore, the focus and content of the research in this manuscript is distinct from previously published papers.

**Technical corrections:**

2. Figs. 1, 3, 4: for the surface reflectance results shown in these figures, which wavelength band is used?

**Response:** In Figures 1, 3, and 4, we utilized the MISR blue band (446 nm) as the data source for surface reflectance. We appreciate the careful review by the reviewer, and we have included this information in the manuscript.

3. Fig. 2: For figure 2, more detailed description needs to be added in section 3.3, such as what kind of aerosol type (model) was used for atmospheric correction and AOD retrieval? Have the authors considered the error transfer caused by the aerosol model and atmospheric correction?

**Response:** We appreciate the guidance and suggestions from the reviewer. In our study, we used continental aerosols for AOD retrieval and atmospheric correction using the 6S model. The selection of an appropriate aerosol type is crucial for obtaining accurate aerosol optical depth. Previous studies have shown that continental aerosols can be used to estimate aerosol optical depth in the Yangtze River Delta region (He et al., 2015). We employed the same aerosol type for AOD retrieval and atmospheric correction and utilized the 6S model for atmospheric correction. Thus, this study did not consider the potential error propagation caused by aerosol type and

atmospheric correction. Based on the reviewer's suggestion, we have added the above content in

section 3.3 of the revised manuscript.

---

## Author Response (AR2)

**Response to the reviewers' comments**

On behalf of the co-authors, we are very grateful to you for giving us the opportunity to revise our manuscript. We really appreciate the reviewers' constructive comments and suggestions on our manuscript, entitled "Multi-angle aerosol optical depth retrieval method based on improved surface reflectance (ID: amt-2023-204) ". We have studied the reviewer's comments carefully and tried our best to revise the manuscript accordingly. Notably, the changes are **highlighted in blue in the revised manuscript.** Please see below for a point-by-point response to the reviewer's comments and concerns.

**Reviewer(s) Comments:**

**Reviewer #1**

**General comments:**

1. The authors have made a lot of changes in response to the reviewer comments. The quality of the manuscript has certainly improved as a result. However, some of my original concerns remain. In my opinion, the improvement in AOD results compared to AERONET values is not very impressive. For example, here are some of the listed R values (old and new) for Taihu:

Da 0.77 -> 0.80, Ca 0.70 -> 0.72, Aa 0.81 -> 0.82

Indeed, in some cases, the results become worse. For example, here are some of the listed RMB values for Taihu:

Ba 0.61 -> 0.70, Aa 0.68 -> 0.71, Df 0.47 -> 0.66

At the very least, the authors need to justify how this is a substantial improvement and how this affects downstream products based on the AOD.

**Response:** Thank you very much for this valuable comment or suggestion. We deeply understand the concerns raised by the reviewer. Because the improvement of the accuracy of AOD retrieval is

not an easy task, we feel that small improvements are also valuable.

As we know, the MISR sensor captures images from 9 different angles within a short span of 7 minutes. Although the time interval is relatively brief, these images are typically considered to be acquired almost simultaneously (Abdou et al., 2005). For features within the same pixel, there is only one shared pixel value. In the final retrieval results, we select the optimal AOD pixel value from the best angle. Prior to the improvement, the retrieval results at angle Aa were optimal (Taihu: R=0.81, RMB=0.68; Xuzhou-CUMT: R=0.73, RMB=0.78). However, after the improvement, overall, the retrieval results at angle An became better (Taihu: R=0.84, RMB=0.52; Xuzhou-CUMT: R=0.85, RMB=0.47). The improvements in the Xuzhou-CUMT site are more pronounced compared to the Taihu site, which may be related to the relatively smaller number of sample points in Taihu. We have also supplemented this part of the content in Section 4.3 of the manuscript. Comparing with MODIS AOD products, the improved AOD retrieval results exhibit better performance in spatial distribution range and trend. The algorithm used in this study refines the spatial distribution of AOD, providing higher spatial resolution and fewer missing pixel values.

In this study, we are dedicated to improving the accuracy and reliability of AOD data, and achieving significant improvements is no easy task. This highlights the challenges and difficulties we face in our research, while also underscoring the importance of ongoing efforts and exploring new methods to achieve more significant improvements. We thank the reviewer for this valuable comments, and we will continue our efforts to actively explore more effective methods to improve the quality and accuracy of AOD retrieval in future.

2. Also, in spite of the changes made, there are still numerous grammatical errors and the language is awkward in several places. For example, inputted is not common usage.

**Response:** We have modified the manuscript including language presentation and logic. These changes do not influence the content and framework of the paper. Here we did not list the changes but marked in blue in the revised manuscript. We appreciate for reviewers' warm work earnestly and hope that the revisions will be recognized.

**Reference**

Abdou, Wedad A. Comparison of coincident multiangle imaging spectroradiometer and moderate resolution imaging spectroradiometer aerosol optical depths over land and ocean scenes containing aerosol robotic network sites[J]. Journal of Geophysical Research, 2005, 110 (D10): 1-12.